# Exploiting Category Names for Few-Shot Classification with Vision-Language Models

## Abstract

Vision-language foundation models pretrained on large-scale data influence many visual understanding tasks. Notably, many vision-language models build two encoders (visual and textual) that can map two modalities into the same embedding space. As a result, the learned representations achieve good zero-shot performance on tasks like image classification. However, when there are only a few examples per category, the potential of large vision-language models is not fully realized, mainly due to the disparity between the vast number of parameters and the relatively limited amount of training data. This paper shows that we can significantly improve the performance of few-shot classification by using the category names to initialize the classification head. More interestingly, we can borrow the non-perfect category names, or even names from a foreign language, to improve the few-shot classification performance compared with random initialization. With the proposed category name initialization method, our model obtains state-of-the-art performance on several few-shot image classification benchmarks (e.g., 87.37% on ImageNet and 96.08% on Stanford Cars, both using five-shot learning). Additionally, we conduct an in-depth analysis of category name initialization, explore the point at which the benefits of category names decrease, examine how distillation techniques can enhance the performance of smaller models, and investigate other pivotal factors and intriguing phenomena in the realm of few-shot learning. Our findings offer valuable insights and guidance for future research endeavors.

## 1 Introduction

In recent years, large vision-language models have opened doors to many new applications and provided new thoughts to existing problems. The advantages of large vision-language models are blessed by learning from largely available images with surrounding texts, as well as exploring the capacity of transformer network (Dosovitskiy et al., 2021) to model web-scale image-text data. Radford et al. (2021) first proposed CLIP for vision-language modeling, which was followed by numerous works, including ALIGN (Jia et al., 2021), LiT (Zhai et al., 2022b), Flamingo (Alayrac et al., 2022), Florence (Yuan et al., 2021), CoCa (Yu et al., 2022), etc. The development of vision-language models provides novel perspectives of few-example learning.

This paper considers the problem of few-shot classification in the new light of large vision-language models. Researchers have found that models pretrained from ImageNet can be easily transferred by finetuning on a new classification task (Huh et al., 2016). Similarly, we can take the vision encoder from the pretrained vision-language model and finetune it with a few examples. Since state-of-the-art vision-language models were pretrained on billions of web images and texts, such finetuning often outperforms the models trained on ImageNet with better robustness and generalization capabilities. Moreover, large vision-language models can be adapted to more downstream tasks with fewer labeled data.

Despite the capability of the text branch in pretrained vision-language models, it is not optimally utilized when directly fine-tuning the vision component for downstream image classification tasks. Additionally, the large size of these models can lead to over-fitting when trained on limited data. In addition to the above approach, we exploit another source of information in vision-language models that traditional models have

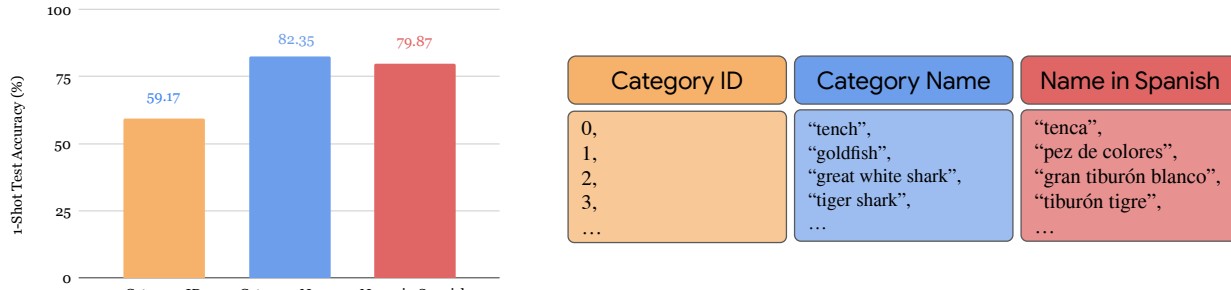

Figure 1: Comparing one-shot classification accuracy on ImageNet using different category information. The typical way of finetuning using images with their category IDs does not work well for one-shot learning with big models. With the information on the category names of training images, we develop a new initialization approach that significantly boosts the performance of vision-language models in few-shot learning. Interestingly, using non-English names can still help even though the model was pre-trained using images and English text data pairs.

overlooked. Such new information comes from the category names in downstream image classification tasks. Because vision-language models can generate powerful representations for images and texts, we will show that by utilizing semantic category names for initialization, vision-language models can be transferred better with few examples in downstream tasks.

As summarized in Figure 1, this paper explores several scenarios: (1) randomly initializing a classification head; (2) initializing a classification head with category names; (3) initializing a classification head with other heuristics such as class digits or even non-English category names. Note that (1) corresponds to the scenario when we only know the category ID (e.g., class 0, class 1, ..., class N) without knowing the meaning of each category. However, (2) implicitly parses the information from category names such as "tench" and "goldfish". The pretrained language model could process these label names to provide a better initialization for the model adaption. Compared to (2), (3) provides different types of category name information. The main difference between scenario (1) and the others is that (1) does not utilize text/language information from the categories. In scenario (1), the backbone network is initialized from the pretrained model weights, and the classification head is randomly initialized. We set (1) to be our baseline as it is the most common model adaptation method. We leverage the pretrained language model for the other scenarios to parse the text information in the provided categories. Specifically, we pair all category names with prompts and extract the average text embedding as the weight to initialize the classification head. The second scenario is called *category name initialization* (CNI), and it has achieved the best performance among all these scenarios when finetuning using one-shot ImageNet data, as shown in Figure 1.

In this paper, we conduct extensive experiments exploring few-shot performance on ImageNet (Deng et al., 2009), Cifar100 (Krizhevsky, 2009), Oxford Flowers (Nilsback & Zisserman, 2008), Stanford cars (Krause et al., 2013), etc. Using the powerful pretrained models, we sweep hyper-parameters such as learning rates, training layers, weight regularization, etc., and find a stable recipe for few-shot learning that can significantly outperform the state-of-the-art in many classification tasks. Notably, we achieve a one-shot top-1 accuracy of 86.15% and a five-shot 87.90% top-1 accuracy on ImageNet, which outperforms many other approaches using the same or more training examples. More interestingly, in this work, we demonstrate that:

- Category name initialization can significantly boost the finetuning performance in few-shot settings, outperforming many other initialization or fine-tuning methods. However, the contribution of category names diminishes when there are a sufficiently large number of training images.

- Leveraging the proposed category name initialization can speed up convergence compared to random initialization.

- In scenarios where a user does not speak English, we find that the non-English category name still helps with few-shot learning. For example, we can use Spanish category names to initialize the network, which is more effective than random initialization.

- A larger pretrained model could further boost the few-shot performance of a small model by carrying out model distillation. We have achieved 1.01% performance boost using 1% labeled images from ImageNet.

- The selection of finetuning layers is crucial to the performance. Empirically, finetuning the last few layers is much better than full model finetuning in a few-shot setting. On the other hand, finetuning the entire network works better when the training data is sufficient.

- We explore additional factors that impact few-shot learning, specifically the learning rate and weight regularization. We provide a comprehensive guide on determining the optimal learning rate and analyze the interesting effects of incorporating $L_2$ weight regularization into few-shot learning.

## 2 Related Work

The human vision system can surprisingly learn from only a few examples. More amazingly, one may learn more effectively by knowing the new species' names. For example, people who have seen "fish" and "cat" before can quickly understand what "catfish" means with or without the help of additional images. Motivated by this phenomenon, few-shot learning has been extensively studied in computer vision (Fei-Fei et al., 2006; Hariharan & Girshick, 2017). Since deep CNN became popular, a common practice is to train a deep CNN on ImageNet and then transfer the model to downstream tasks (Huh et al., 2016). However, transferring a pretrained ImageNet model requires hundreds or thousands of images. When there are only a few examples per category, the few-shot learning using pretrained ImageNet models is inferior to those trained with enough in-domain data.

Recently, there has been increasing interest in utilizing the vision-language model for visual zero-shot learning, a related problem of few-shot learning. CLIP (Radford et al., 2021) is a pioneering work in large-scale vision-language modeling. Unlike previous works in vision-language representation (Donahue et al., 2015; Vinyals et al., 2015), CLIP collects image-text pairs from the Web, which contains diversified semantics in a weakly supervised fashion. In addition, CLIP is built on large-scale contrastive learning, which maps images and text into the same subspace. Through this, the model can map textual class names with images hence performing image classification in a zero-shot manner. The approach of CLIP was followed by ALIGN (Jia et al., 2021), Flamingo (Alayrac et al., 2022), LiT (Zhai et al., 2022b), Florence (Yuan et al., 2021), FLAVA (Singh et al., 2022), SimVLM (Wang et al., 2022) and CoCa (Yu et al., 2022). Among these works, ALIGN, Florence, FLAVA, and LIT are based on contrastive learning. Flamingo chooses to optimize a generative loss with gated cross-attention layers. At last, CoCa integrates contrastive and generative loss into one framework. Although training CoCa seems the most challenging among all these vision-language works, it obtains consistently better results in many tasks.

In the literature, CLIP, LiT, ALIGN, Florence, FLAVA, and CoCa have demonstrated promising results with zero-shot learning. However, the potential of these models for few-shot learning is not well exploited. Li et al. (2022) construct a benchmark and toolkit named Elevater for evaluating the transferability of vision-language models using different training samples. Radford et al. (2021) point out that using few training examples could improve the effectiveness robustness while undermining the relative robustness. Few-shot learning algorithms are trained exclusively on image data, ignoring the valuable text information that can be used to enhance the learning process. However, Flamingo has emerged as a promising approach for addressing this issue. Flamingo utilizes few-shot interleaved prompts that incorporate gated cross-attention layers to improve few-shot learning.

Zhou et al. (2022a) propose context optimization (CoOp) to model text in prompts through continuous representations. Zhou et al. (2022b) propose CoCoOp, which extends CoOp by further learning a lightweight neural network to generate an input-conditional token (vector) for each image. In addition, a series of prior-based methods utilize CLIP priors with a cache model. CLIP-Adapter (Gao et al., 2021) combines

zero-shot visual or language embeddings with corresponding finetuning features to improve performance. TIP-Adapter (Zhang et al., 2022) constructs adapters using a key-value cache model from few-shot training sets and updates their prior knowledge through feature retrieval. TIP-X (Udandarao et al., 2022) further constructs an affinity matrix by measuring the KL divergence between test and few-shot samples, which removes direct reliance on the uncalibrated image-image similarities. APE (Zhu et al., 2023) explores the trilateral affinities between the test image, prior cache model, and textual representations and only enables a lightweight category-residual module to be trained. Among these approaches, TIP-Adapter, TIP-X, and APE are training-free, while CoOp, CoCoOp, CLIP-Adapter, and APE-T (Zhu et al., 2023) require training. Klein et al. (2014) suggest that using a fisher vector derived from other distributions can improve accuracy in central computer vision tasks. Category names have also been exploited in image-text tasks, such as visual grounding (Wang et al., 2017) and visual question answering (Gupta et al., 2017). In these methods, the text embedding of the category names and the image embedding is extracted separately by two branches. Then their inner product is calculated as the similarity score between an image region and an object category.

This paper demonstrates that leveraging category names for initialization can significantly enhance the few-shot performance of the CoCa model without bells and whistles. Our approach outperforms Flamingo and CLIP's performance and establishes a new state-of-the-art for both ImageNet and several other datasets with fewer training examples.

## 3  Approach

In this section, we first briefly review CoCa (Yu et al., 2022), one of the state-of-the-art vision-language models, and then discuss two initialization strategies: the standard random initialization and new category name initialization (CNI) for finetuning tasks.

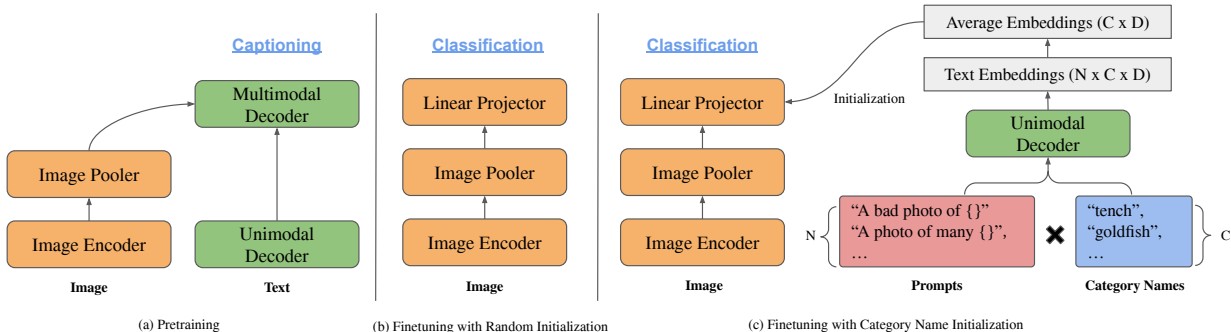

Figure 2: An overview of the CoCa pretraining and finetuning. (a) The pretraining of CoCa relies on mapping image and text pairs into the same space for embedding alignment, where the image and text embeddings are extracted through an image encoder and a unimodal text decoder, respectively. The image pooler is used to customize the image embedding for different tasks. (b) We append a randomly initialized linear projector to the image pooler and initialize the image encoder from pretrained weights. (c) We construct text sequences by pairing all $C$ category names with $N$ different prompts. Via the pretrained unimodal decoder, we can compute the text embeddings for all text sequences (with a total number of $N \times C$), each of which is a $D$-dimensional vector. The normalized average embeddings can be used to initialize the linear projector's weight.

### 3.1  Revisiting CoCa pretraining

Unlike other recent vision-language models, CoCa adopts an encoder-decoder model architecture to learn the generic vision and multi-modal representations. As shown in Figure 2 (a), CoCa encodes images to latent representation via an encoder network (e.g., vision transformer (ViT) (Dosovitskiy et al., 2021)) and encodes text representations via a unimodal decoder. We append an image pooler after the image encoder to customize the image representations. Practically, CoCa adopts a cascade design by using two

image poolers, i.e., a generative image pooler and a contrastive image pooler. The motivation for this design comes from the preliminary experimental results that single pooled image embedding helps vision recognition tasks while more visual tokens benefit multi-modal understanding tasks. Following Lee et al. (2019), both generative and contrastive image poolers are single multi-head attention layers with different numbers of learnable queries, enabling the model to pool embedding with different lengths. They can also customize visual representations for different tasks and training objectives. For simplicity and clarity, we depict them in one box named *Image Pooler*. On the other hand, CoCa uses a unimodal decoder to extract text-only embeddings. It cascades multi-modal decoder layers cross-attending to image embeddings to learn multi-modal image-text representations.

CoCa is pretrained on image-text pairs using two objective functions. The first is contrastive loss, where the image representations are contrasted against the paired text representations. The contrastive loss enables cross-modal representation alignment. The other is image-captioning loss, which requires the model to auto-regressively predict the tokenized texts by maximizing the conditional likelihood. The resulting CoCa can thus generate both unimodal visual/textual embeddings and multi-modal joint embeddings. The unimodal visual output generated by the encoder and the unimodal textual output generated by the unimodal decoder are aligned in the same vector space and thus can be used to map images with their class names in a zero-shot manner. Here, we focus on reusing these two components to initialize for few-shot learning.

### 3.2 Finetuning CoCa

**Random initialization.** One straightforward model adaption approach is to add a randomly initialized linear projector upon the pretrained model and selectively finetune the model (all or part of the layers), as depicted in Figure 2 (b). Following the approach used by CLIP (Radford et al., 2021) and CoCa (Yu et al., 2022), we first use an image pooler to obtain the aggregated image embedding $H \in \mathcal{R}^D$ and then apply a linear projector to get the prediction $Y \in \mathcal{R}^C$,

$$Y = \mathrm{softmax}(WH + b), \tag{1}$$

where $W \in \mathcal{R}^{C \times D}$ and $b \in \mathcal{R}^C$ are learnable weight and bias of the linear projector. Here $W$ and $b$ are randomly initialized, while the image encoder and generative image pooler are initialized from the pretrained weights. Table 7 summarizes the number parameters of different modules of CoCa.

**Category name initialization.** We argue that the above random initialization ignores the potential of the language model for model adaptation. In contrast, we propose the category name initialization to maximize the capacity of the pretrained unimodal decoder. First, we pair all category names (whose total number is $C$) with $N$ different prompts as the text inputs. For example, pairing the category name "tench" with a prompt "A bad photo of {}" gives us a text sequence "A bad photo of tench". Next, we compute the text embeddings for all these $N \times C$ text sequences via the unimodal decoder. As the text embedding for each text input is a $D$-dimensional vector, we can obtain a text embedding tensor with a shape of $N \times C \times D$. Following the previous work CLIP (Radford et al., 2021), we compute the average over different prompts and perform the $L_2$ normalization to obtain the average embeddings of shape $C \times D$. Unlike random initialization, we initialize the weight $W$ by the average embeddings and bias $b$ by a zero vector in the linear projector. We initialize the image encoder and the image pooler from the pretrained model weights to enable zero-shot inference of the category name initialized model.

**Discussion.** Category name initialization is model-agnostic, making it applicable to other foundation models that utilize contrastive loss. Vision-language models trained with contrastive learning inherently yield a two-tower representation, where the text tower's output is embedded into the image tower's embedding space. This shared embedding space allows for cosine distance computation through the inner product of normalized embedding vectors. Consequently, the text embeddings of category names can effectively initialize the visual classifier.

With category name initialization, the model can maintain its zero-shot performance even before fine-tuning, avoiding starting from scratch and undergoing a lengthy fine-tuning process. In contrast to CoOp (Zhou et al., 2022a), where prompts are learnable variables, the prompts for each downstream image classification

task are fixed. In Section 4.3, we will demonstrate that context optimization is less effective than category name initialization. TIP-Adapter (Zhang et al., 2022) calculates predicted logits by measuring the affinity between embeddings of the test image and cached training images, as well as textual embeddings. In Section 4.4, we will show that using cached image embeddings for initialization leads to poorer performance. CLIP-Adapter (Gao et al., 2021) introduces and fine-tunes two learnable adapters, each consisting of two layers of linear transformations, to transform classifier weights and image features. However, we found that using text embeddings of category names to initialize the classifier and finetuning the final few layers is the most effective method for few-shot learning without the need for overly complex designs. Layer selection will be discussed in Section 4.7.

In practice, we may not always have the names of all categories. For example, when the finetuning service is provided to users from another country with different languages, the user may use category names in a foreign language or even digital labels for each category. Interestingly, although trained only with English texts, CoCa uses a word piece model and sentence piece model as the tokenizer and thus can compute the embedding for any text sequence without reporting the out-of-vocabulary error. In Section 4.4, we will compare the impact of different variants of category name initialization.

## 4 Experiments

In this section, we first describe the details of our experimental setups, and then present our experimental results as well as key findings with comprehensive analysis.

### 4.1 Experimental Setup

**Data.** We conducted finetuning experiments on several widely-used image classification datasets, including ImageNet (Deng et al., 2009), ImageNet-V2 (Recht et al., 2019), ImageNet-R (Hendrycks et al., 2021a), ImageNet-A (Hendrycks et al., 2021b), ImageNet-Sketch (Wang et al., 2019), Cifar100 (Krizhevsky, 2009), Oxford Flowers (Nilsback & Zisserman, 2008), Stanford Cars (Krause et al., 2013), Country-211 (Radford et al., 2021), Food-101 (Bossard et al., 2014), FGVC Aircraft (Maji et al., 2013), EuroSAT (Helber et al., 2019), and Oxford-IIIT Pets (Parkhi et al., 2012). To account for different few-shot settings, we randomly sampled a specific portion of data from each dataset. For instance, in one-shot ImageNet, we only chose one image from the ImageNet training data for each category. Despite this sampling, we evaluated all models on the entire testing set. Following the existing benchmark (Li et al., 2022), we employed the same text prompts[1] for evaluating all methods for a fair comparison. CoCa (Yu et al., 2022) is pretrained using JFT-3B (Zhai et al., 2022a) and Align datasets (Jia et al., 2021). During the pretraining stage, all near-domain examples (3.6M images) are removed following the strict de-duplication procedures (Zhai et al., 2022a; Jia et al., 2021).

**Optimization.** We use the Adafactor optimizer (Shazeer & Stern, 2018) with $\beta_1 = 0.9$, $\beta_2 = 0.999$, and a weight decay ratio of 0.01. All input images are first rescaled to $580 \times 580$ and then randomly cropped to the size of $540 \times 540$. We further apply RandAugment (Cubuk et al., 2020) and label smoothing in our data preprocessing pipeline. Our model is implemented in the Lingvo framework using Tensorflow (Shen et al., 2019).

**Hyper-parameters.** The choice of batch size depends on the dataset and its number of categories. When the total number of training examples is relatively small, using a large batch size may not be feasible. However, using the largest possible batch size for efficient training is generally desirable. For instance, in the case of ImageNet, which consists of 1000 categories, we opt for a batch size of 512. This decision is based on the consideration that we have a substantial number of images per category, either 1000 images (for one-shot tasks) or 5000 images (for five-shot tasks). Therefore, using a batch size of 512, we can efficiently utilize the available computational resources during training. However, it is important to note that the batch size is adjusted accordingly for datasets with a smaller number of categories. For instance, in the case of Cifar-100,

---

[1] https://github.com/Computer-Vision-in-the-Wild/Elevater_Toolkit_IC/blob/main/vision_benchmark/datasets/prompts.py

Table 1: Few-shot results on ImageNet and its variants. We use IN as the abbreviation for ImageNet, and CNI for category name initialization. The second column means how much training data per class is used for finetuning. 0 shot means the pretrained vision-language model is directly evaluated without finetuning. Full means the entire training set has been used. All the numbers under the last five columns denote the top-1 test accuracy.

| Model | Shot | IN | IN-V2 | IN-R | IN-A | IN-Sketch |
|---|---|---|---|---|---|---|
| MAE (He et al., 2022) | full | - | - | 66.50 | 76.70 | 50.90 |
| CLIP (ViT-B/16) (Radford et al., 2021) | 0 | 68.40 | 62.60 | 77.60 | 50.00 | 48.20 |
| | full | 79.90 | 69.80 | 70.80 | 46.40 | 46.90 |
| CLIP (ViT-L/14) (Radford et al., 2021) | 0 | 76.20 | 70.10 | 88.90 | 77.2 | 60.20 |
| | full | 85.20 | 75.80 | 85.30 | 76.10 | 58.70 |
| CLIP+Adapter (ResNet-50) (Gao et al., 2021) | 0 | 55.50 | - | - | - | - |
| | 1 | 58.10 | - | - | - | - |
| | 4 | 59.50 | - | - | - | - |
| CLIP+CoOp (ViT-B/16) (Zhou et al., 2022a) | 0 | 58.18 | - | - | - | - |
| | 1 | 58.00 | - | - | - | - |
| | 4 | 60.01 | - | - | - | - |
| Tip-Adapter-F (ResNet-50) (Zhang et al., 2022) | 0 | 60.33 | - | - | - | - |
| | 1 | 61.32 | - | - | - | - |
| | 4 | 62.52 | - | - | - | - |
| WiSE-FT (ViT-L/14) (Wortsman et al., 2022) | full | 85.30 | 76.90 | 89.80 | 79.70 | 63.00 |
| Flamingo-3B (Alayrac et al., 2022) | 1 | 70.90 | - | - | - | - |
| | 5 | 72.70 | - | - | - | - |
| Flamingo-80B (Alayrac et al., 2022) | 1 | 71.90 | - | - | - | - |
| | 5 | 77.30 | - | - | - | - |
| CoCa-base (Yu et al., 2022) | 0 | 82.26 | 76.22 | 93.16 | 76.17 | 71.12 |
| CoCa-base+CNI (Ours) | 1 | 82.35 | 76.47 | 93.37 | 77.00 | 71.61 |
| | 5 | 83.58 | 77.23 | 93.22 | 77.23 | 71.35 |
| CoCa-2B (Yu et al., 2022) | 0 | 86.09 | 80.39 | 96.19 | 89.39 | 77.12 |
| CoCa-2B+CNI (Ours) | 1 | 86.15 | 80.57 | 96.62 | 90.12 | 77.49 |
| | 5 | 87.37 | 81.66 | 96.41 | 89.68 | 77.39 |

where there are 100 categories, we choose a batch size of 256 for the five-shot setting and 64 for the one-shot setting.

**Model cost.** The computational cost of training a model depends on the model size and the chosen training batch size. To provide specific examples, when fine-tuning CoCa-base on ImageNet (five-shot), we utilized a 4x4 Jellyfish TPU with a batch size of 512, and the training process took approximately 6 hours. Similarly, when fine-tuning CoCa-2B on Cifar-100 (five-shot), we employed a 4x4 Dragonfish TPU with a batch size 256, and the training duration was around 9 hours.

### 4.2 Improving CoCa in few-shot classification

**State-of-the-art on ImageNet and its variants.** We use the pretrained CoCa model and apply category name initialization. We then compare our method against the previous works on ImageNet and its variants, including ImageNet-V2 (Recht et al., 2019), ImageNet-R (Hendrycks et al., 2021a), ImageNet-A (Hendrycks et al., 2021b) and ImageNet-Sketch (Wang et al., 2019). As shown in Table 1, CoCa-2B+CNI has achieved state-of-the-art few-shot classification results on all these benchmarks. Surprisingly, the one-shot and five-

Table 2: Comparing with the state-of-the-art on multiple classification benchmarks. CNI stands for category name initialization, and RI means random initialization. Our model obtains the state-of-the-art few-shot learning performance with less training data than others.

| Model | Shot | Cifar100 | Oxford Flowers | Stanford Cars | Country-211 | Food-101 | FGVC Aircraft | EuroSAT | Oxford-IIIT Pets |
|---|---|---|---|---|---|---|---|---|---|
| MAE (He et al., 2022) | 5 | 21.20 | 50.90 | 6.30 | 2.80 | 7.70 | 7.00 | 64.60 | 17.20 |
| | 20 | 43.50 | 71.90 | 25.50 | 4.40 | 30.40 | 29.90 | 74.10 | 60.00 |
| | full | 68.30 | 72.00 | 37.20 | 10.10 | 65.10 | 39.10 | 94.80 | 81.60 |
| CAE (Chen et al., 2022) | 5 | 38.30 | 70.30 | 8.70 | 3.50 | 18.60 | 14.30 | 76.70 | 37.30 |
| | 20 | 55.10 | 81.20 | 27.50 | 5.50 | 35.70 | 32.60 | 89.00 | 63.30 |
| | full | 78.90 | 81.20 | 40.40 | 11.40 | 67.40 | 40.80 | 96.70 | 79.80 |
| MoCo-v3 (Chen et al., 2021) | 5 | 60.50 | 79.50 | 13.40 | 4.80 | 36.60 | 11.80 | 77.10 | 76.20 |
| | 20 | 75.50 | 89.50 | 49.50 | 7.60 | 59.30 | 38.20 | 84.80 | 86.40 |
| | full | 85.30 | 89.50 | 63.00 | 13.70 | 78.00 | 48.00 | 95.90 | 91.40 |
| DeiT (Touvron et al., 2021) | 5 | 61.50 | 82.70 | 27.60 | 4.40 | 41.90 | 24.10 | 62.50 | 87.80 |
| | 20 | 73.70 | 92.70 | 68.80 | 6.20 | 61.50 | 34.10 | 90.70 | 91.90 |
| | full | 89.60 | 92.40 | 83.00 | 14.10 | 84.50 | 59.30 | 98.20 | 93.90 |
| ViT (Dosovitskiy et al., 2021) | 5 | 75.40 | 99.20 | 27.60 | 6.80 | 59.00 | 22.70 | 70.00 | 89.60 |
| | 20 | 84.00 | 99.20 | 53.90 | 11.50 | 81.70 | 40.50 | 86.50 | 92.60 |
| | full | 89.80 | 99.20 | 67.50 | 16.60 | 89.60 | 47.80 | 96.00 | 94.80 |
| CLIP (Radford et al., 2021) | 5 | 71.10 | 94.20 | 73.60 | 21.70 | 89.70 | 36.00 | 76.70 | 90.50 |
| | 20 | 75.40 | 96.8 | 73.60 | 25.20 | 90.60 | 48.10 | 86.60 | 92.30 |
| CoCa-2B (Yu et al., 2022) | 0 | 77.19 | 92.04 | 94.37 | 42.15 | 94.79 | 44.83 | 49.74 | 97.88 |
| CoCa-2B+RI | 1 | 5.69 | 40.78 | 14.29 | 1.71 | 1.26 | 12.24 | 56.84 | 61.95 |
| | 5 | 7.49 | 84.71 | 86.31 | 19.06 | 62.45 | 27.21 | 82.38 | 78.61 |
| CoCa-2B+CNI | 1 | 77.89 | 98.45 | 95.29 | 42.44 | 94.91 | 58.33 | 75.06 | 97.93 |
| | 5 | 78.62 | 99.25 | 96.08 | 44.52 | 95.50 | 69.29 | 85.78 | 98.12 |

shot performance of CoCa-base is even better than the performance of some other recent methods finetuned on the whole dataset.

**State-of-the-art on other benchmarks.** In addition to ImageNet and variants, we show that our method can achieve state-of-the-art few-shot performance on other image classification benchmarks, including Cifar100 (Krizhevsky, 2009), Oxford Flowers (Nilsback & Zisserman, 2008) and Stanford Cars (Krause et al., 2013), Country-211 (Radford et al., 2021), Food-101 (Bossard et al., 2014), FGVC Aircraft (Maji et al., 2013), EuroSAT (Helber et al., 2019), and Oxford-IIIT Pets (Parkhi et al., 2012). By examining Table 2, it becomes apparent that our CoCa-2B model outperforms many other approaches, even when trained with fewer data. The performance gain results from the category name initialization, which serves as a strong foundation that enables the model to achieve better results with only a few examples. To gain a deeper understanding of this phenomenon, we provide an analysis of the category name initialization in the following section.

### 4.3 Analysis of category name initialization

This section delves deeper into how the proposed category name initialization helps with large vision-language models in few-shot learning. Vision-language models are adept at zero-shot inference without knowing any class names from downstream tasks. However, the zero-shot performance heavily depends on the domain gap and data distribution, thus varying on different downstream tasks. By leveraging a few training examples from the target domain, the pretrained vision-language models can adapt to the target domain.

**Improvement upon zero-shot performance.** We first examine how category name initialization improves zero-shot performance. As illustrated in Table 1 and Table 2, category name initialization enhances performance across all datasets. The improvement in performance from zero-shot to five-shot varies depending on the dataset. For instance, CoCa-2B on ImageNet sees a 1.32% increase in performance, whereas EuroSAT sees 36.04% growth. CoCa's zero-shot performance on ImageNet leaves less room for few-shot learning. Nonetheless, the performance gain achieved through our category name initialization is notewor-

Table 3: Comparing other fine-tuning methods on ImageNet and its variants. We use IN as the abbreviation for ImageNet, and CNI for category name initialization. The second column means how much training data per class is used for finetuning. 0 shot means the pretrained vision-language model is directly evaluated without finetuning. All the numbers under the last five columns denote the top-1 test accuracy.

| Model | Shot | IN | IN-V2 | IN-R | IN-A | IN-Sketch |
|---|---|---|---|---|---|---|
| CoCa-base | 0 | 82.26 | 76.32 | 93.16 | 76.17 | 71.43 |
| CoCa-base+Linear Probing | 1 | 57.49 | 54.20 | 69.19 | 53.38 | 47.94 |
| | 5 | 79.33 | 73.18 | 90.02 | 73.18 | 68.03 |
| CoCa-base+Full Fintuning | 1 | 43.77 | 41.64 | 55.98 | 40.31 | 33.29 |
| | 5 | 60.90 | 54.32 | 71.20 | 54.34 | 49.25 |
| Coca-base+CoOp | 1 | 79.85 | 73.21 | 89.88 | 76.42 | 65.81 |
| | 5 | 81.01 | 75.81 | 92.58 | 76.55 | 71.27 |
| CoCa-base+CNI | 1 | 82.35 | 76.47 | 93.37 | 77.00 | 71.61 |
| | 5 | 83.47 | 77.23 | 93.22 | 77.23 | 71.35 |

thy, as some other methods may not achieve comparable improvements, which will be discussed below. We also contend that our few-shot performance is not solely attributable to the strong pretrained CoCa model but also our proposed category name initialization. For example, CoCa-2B's zero-shot performance on EuroSAT is 49.74%, which is lower than that of most other approaches. However, with our category name initialization, it achieves 85.78%, outperforming other approaches in the five-shot setting.

**Comparing with other fine-tuning methods.** To further validate the efficacy of category name initialization, we compare it with several other finetuning methods. We choose CoCa-base as the pretrained vision-language model and carry out experiments on ImageNet with different finetuning methods, such as linear probing, full finetuning, CoOp (Zhou et al., 2022a), and category name initialization. As demonstrated in Table 3, all finetuning methods, except category name initialization, fail to improve over zero-shot CoCa when one or five training examples per class are used. Furthermore, full finetuning underperforms linear probing because the number of training examples is inconsistent with the number of trainable parameters in few-shot learning. Although showing better performance than linear probing and full finetuning, the one- or five-shot performance of CoOp is slightly inferior to zero-shot CoCa. This suggests that learning contextual prompts does not significantly improve CoCa's few-shot performance. On the other hand, category name initialization effectively improves the few-shot performance, which is challenging when the zero-shot performance of CoCa is significantly higher than that of other counterparts such as CLIP (Radford et al., 2021) and FLAVA (Singh et al., 2022).

**Category name initialization vs. random initialization.** To gain a deeper understanding of the advantages of category name initialization, we compared it with random initialization. Comparing the last three rows in Table 2, we can observe that the few-shot classification results using random initialization are worse than the zero-shot classification with pretrained CoCa. However, employing category name initialization would effectively use those few training examples and boost performance. Figure 3 provides a more detailed comparison of the optimization process using the two initialization methods. By meticulously tuning the parameters, we set the initial learning rate to 1e-5 for category name initialization and 5e-5 for random initialization. Employing category name initialization results in a better starting model with higher test accuracy than random initialization. Furthermore, the model utilizing category name initialization converges faster than random initialization. This can be attributed to the fact that the test accuracy while using random initialization continues to increase even after 250 epochs, whereas the accuracy achieved with category name initialization plateaus around 200 epochs when fine-tuning on ImageNet. Similarly, the one-shot test accuracy on Cifar-100 converges within 100 epochs by employing category name initialization, while the counterpart using random initialization converges after 300 epochs.

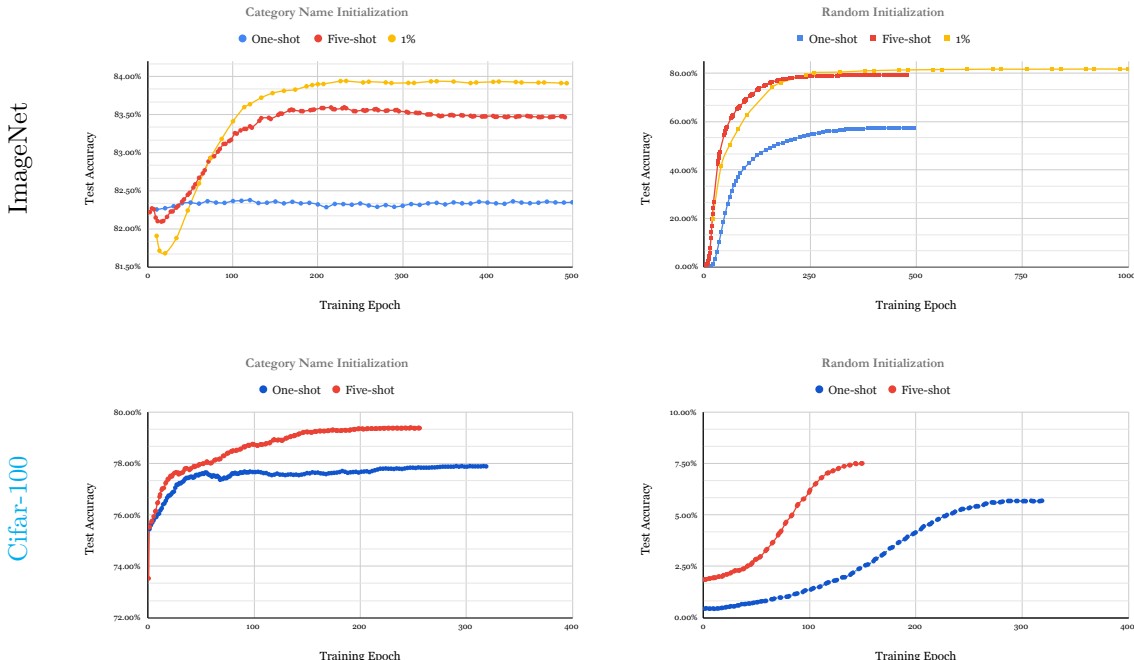

Figure 3: Comparison of test accuracy over the training epoch. We finetune the CoCa-base model with category name initialization or random initialization. Category name initialization provides better initial test accuracy and helps the model converge better and faster than random initialization.

## 4.4 Exploring different initialization approaches

In real-world scenarios, we cannot always guarantee the availability of perfect category names for every classification task. Sometimes we may only have digital labels such as class "1", "2", and so on, while in other cases, users may not be fluent in English. In such scenarios, it is crucial to evaluate how the model performs with different versions of category names.

Table 4 compares the performance of using no category names (i.e., random initialization) with various variants of category names. The most straightforward approach is to use digits (class 1, 2, and so on) as category names. However, this approach provides little semantic information and does not improve few-shot performance. Conversely, category names in English and other languages significantly enhance few-shot recognition. This is surprising because CoCa was trained on English-only text with limited knowledge of other languages. Nevertheless, due to the sentence piece tokenizer (Kudo & Richardson, 2018) and token sharing, our method can still benefit from foreign language transfer, resulting in better performance than random initialization, even though the performance of these foreign language names is not as good as that of English names.

Inspired by the aforementioned observation, we hypothesize that initialization with only partial category information can still yield benefits. To test this hypothesis, we randomly selected 50% of the category names for initialization while using random initialization for the remaining names. The results are shown in Table 5, where it can be seen that using 50% of the names still improves the one-shot accuracy from random initialization from 59.17% to 66.82%, and the five-shot accuracy from 79.33% to 80.67%. This indicates that our method has the potential as a valuable tool in situations where within-domain labels are incomplete or expressed in different languages.

Another question that arises is whether we can apply a similar initialization approach using image embeddings instead of text. To test this hypothesis, we select one representative image per class from ImageNet, resulting in 1000 images for 1000 categories, and used the pretrained CoCa-base model to extract 1000 embedding

Table 4: Comparison of category name initialization using digits or different languages. We use the same pretrained CoCa-base model for all category name initialization. The numbers below are top-1 test accuracy on ImageNet.

| Category Name Initialization | Zero-shot | One-shot | Five-shot |
|:---:|:---:|:---:|:---:|
| No | N/A | 59.17 | 79.33 |
| Digits | 0.10 | 53.60 | 78.75 |
| Korean | 22.89 | 53.71 | 79.53 |
| Russian | 43.59 | 53.43 | 79.55 |
| Germany | 29.24 | 63.15 | 79.90 |
| Spanish | 34.38 | 79.87 | 80.05 |
| English | 82.26 | 82.35 | 83.58 |

Table 5: Comparing the performance of using all category names or 50% of names (the other half will be initialized with random vectors) for initialization. The numbers below are top-1 test accuracy on ImageNet.

| Initialization | Zero-shot | One-shot | Five-shot |
|:---:|:---:|:---:|:---:|
| No category name | N/A | 59.17 | 79.33 |
| 50% category names | 44.36 | 66.82 | 80.67 |
| 100% category names | 82.26 | 82.35 | 83.58 |

vectors. We then initialize the linear projector of our few-shot model with these image embeddings, which we call image embedding initialization (IEI). We compare the performance of IEI (using one example image per category) with CNI (using category names but no images) and present the accuracy of initialized models (without finetuning) in Table 6. The results indicate that IEI performs worse than CNI, suggesting that embedding category names are more robust than embedding a single image. Moreover, we compute the average of the IEI and CNI weights to create a new initialization vector and find that the average weight's performance lies in the middle of IEI and CNI.

Table 6: Comparing top-1 accuracy of image embedding initialization (IEI) and category name initialization (CNI) on ImageNet.

| Initialization | Accuracy (%) |
|:---:|:---:|
| IEI | 47.16 |
| $0.5 \times$ IEI + $0.5 \times$ CNI | 61.84 |
| CNI | 82.26 |

### 4.5 Limitations

After comparing different initialization approaches, one question that arises is whether category name initialization continues to be helpful with more training data. We investigate this by fine-tuning pretrained vision-language models using varying numbers of training images. To demonstrate the effectiveness, we establish a baseline for comparison by using random initialization. We utilize two different pretrained CoCa models, CoCa-base and CoCa-2B, and fine-tune them on ImageNet and Cifar100 using different training data. As shown in Figure 4, category name initialization outperforms random initialization across different datasets, model architectures, and numbers of training data. However, the contribution of category name initialization diminishes as more training data is provided.

Another limitation of the proposed category name initialization is that it relies on category names to initialize the classification head. While it can significantly improve few-shot image classification accuracy, it may not

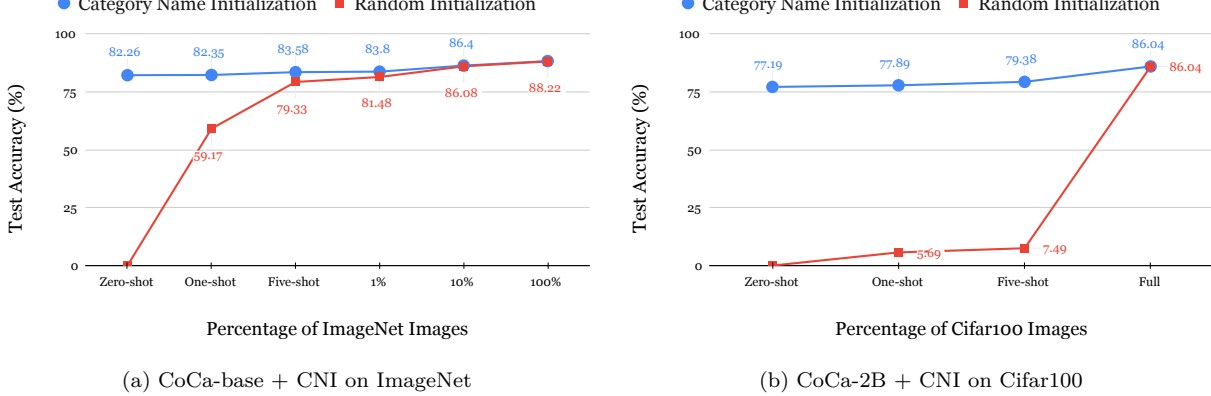

(a) CoCa-base + CNI on ImageNet          (b) CoCa-2B + CNI on Cifar100

Figure 4: Comparison of test accuracy over different percentages of training images. Category name initialization outperforms random initialization over different datasets, model architectures, and numbers of training data.

be applicable in all scenarios. For example, in domains where category names are not available or are not reliable, the proposed method may not be effective.

## 4.6 Model distillation

We first show that category name initialization can be used for different scales of models by carrying out few-shot experiments using two different pretrained CoCa architectures: CoCa-base and CoCa-2B, under different numbers of training data. Abandoning the uni-modal and multi-modal text decoders, CoCa-base and CoCa-2B contain 96M and 1B parameters for downstream image classification tasks (see Table 7). As shown in Table 8, we can observe the trend that bigger models do better and more shots help.

Table 7: Number of parameters of different modules.

Table 8: Few-shot results of different CoCa-models on ImageNet.

| Module | CoCa-base | CoCa-2B |
|---|---|---|
| Image encoder | 85,999,872 | 1,011,740,288 |
| Image pooler | 19,095,296 | 63,843,648 |
| Linear projector | 769,000 | 1,409,000 |

| Model | Zero-shot | One-shot | Five-shot | 1% |
|---|---|---|---|---|
| CoCa-2B | 86.19 | 86.15 | 87.37 | 87.90 |
| CoCa-base | 82.26 | 82.35 | 83.58 | 83.80 |
| + distillation | - | - | - | 84.81 |

As larger models tend to perform better, it is natural to consider knowledge distillation, which involves using the predictions of a teacher model to guide the training of a student model. In this work, we use the finetuned CoCa-2B model with 1% of the ImageNet images as the teacher model and CoCa-base as our student model. In addition to the 1% labeled ImageNet images, we use other unlabeled images for knowledge distillation. During the finetuning process, we freeze the teacher model weights and update the student model weights using two loss objectives. The first objective is the supervised loss, where we compute the cross entropy between the student model predictions and the labels for the 1% labeled ImageNet images. The second objective is the distillation loss, computed over all unlabeled data. Unlike few-shot finetuning, where only the last few layers are finetuned, we finetune the entire student model here since the distillation loss is computed over many unlabeled images. For example, table 8 shows that by distilling from the larger finetuned teacher model, CoCa-base achieves a 1.01% improvement in accuracy (from 83.80% to 84.81%).

## 4.7 Ablation studies

In this section, we analyze several important factors that influence the few-shot performance. We conduct our ablation study using CoCa-base as the model.

**Finetuning layers.** We evaluate the performance of the CoCa-base model on ImageNet (Deng et al., 2009) in various few-shot learning scenarios, with different finetuning layers selected. We compare the results to a baseline using random initialization. In our notation, P denotes the image pooler and L denotes the linear projector. For both category name initialization and random initialization, we experiment with three different optimization strategies: 1) optimizing only the linear projector (L); 2) optimizing both the image pooler (P) and the linear projector (L); and 3) optimizing all layers. Note that we have extensively tried various hyper-parameters (such as initial learning rate) and presented the optimal values for each setting.

The results presented in Table 9 indicate that the best performance is achieved by finetuning both the image pooler and linear projector under all settings when compared to the other two optimization strategies for random initialization.

To enhance the few-shot learning performance, we experiment with category name initialization discussed in Section 3.2. In contrast to random initialization, we initialize the linear projector using the average text embeddings of the category names. As shown in Table 10, this initialization method significantly improves few-shot recognition performance. Moreover, we observe that finetuning P + L is the most effective optimization strategy for few-shot settings while finetuning all layers performs better with more training data.

Table 9: Comparison of different finetuning layers for random initialization. P: image pooler; L: linear projector; All: all layers. The best performance of each column is in **bold**.

| Finetuning Layers | One-shot | Five-shot | 1% | 100% |
|---|---|---|---|---|
| L | 49.38 | 69.64 | 76.53 | 85.62 |
| P + L | **57.49** | **79.33** | **81.48** | **88.22** |
| All | 43.77 | 60.90 | 79.75 | 86.03 |

Table 10: Comparison of different finetuning layers for category name initialization. P: image pooler; L: linear projector; All: all layers. The best performance of each column is in **bold**.

| Finetuning Layers | One-shot | Five-shot | 1% | 100% |
|---|---|---|---|---|
| L | 82.35 | 81.03 | 81.67 | 86.16 |
| P + L | **82.35** | **83.58** | **83.91** | 88.25 |
| All | 82.28 | 82.63 | 83.63 | **88.35** |

**Learning rates.** We analyze the influence of the initial learning rate on few-shot learning. We set a batch size of 512, froze the image encoder, and adopted a cosine learning rate schedule for the final three layers. Figure 5 presents the top-1 test accuracy on ImageNet using different initial learning rates. A small initial learning rate (5e-6) results in a slow convergence rate, while a larger learning rate (5e-5) achieves faster convergence. However, despite reaching the highest test accuracy within 1000 training steps, the finetuning becomes unstable as the test accuracy declines right after the peak value. Conversely, using an even larger learning rate (5e-4) could prevent the surging phase, resulting in a downward trend of test accuracy. By contrast, selecting an appropriate learning rate (1e-5) is the key to stable and rapid few-shot finetuning. Unfortunately, there is no mathematical formula for determining the optimal initial learning rate since it varies across different datasets and depends on the batch size. We can adjust the initial learning rate by trial and observation, and these four test accuracy curves could indicate whether to enlarge or reduce the initial learning rate.

$L_2$ **weight regularization.** Out of all the few-shot settings, one-shot learning is the most unique and intriguing. As illustrated in Figure 6, the one-shot test accuracy (in red) on ImageNet decreases even with category name initialization during finetuning, unlike the five-shot accuracy (in blue). Using only one training image per class can easily distort the decision boundary, as illustrated in Figure 7. We plot the decision boundary in Figure 7 for an illustration. Without $L_2$ regularization, the decision boundary of the

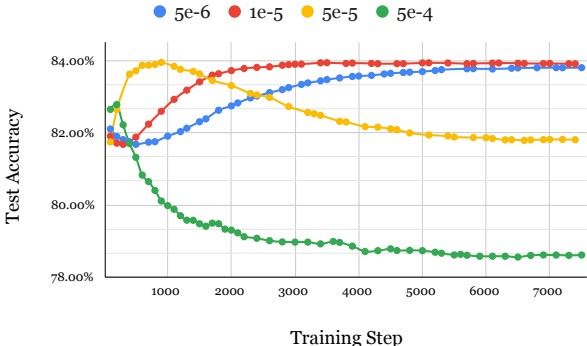

Figure 5: The top-1 test accuracy of finetuning CoCa-base on 1% ImageNet using different initial learning rates.

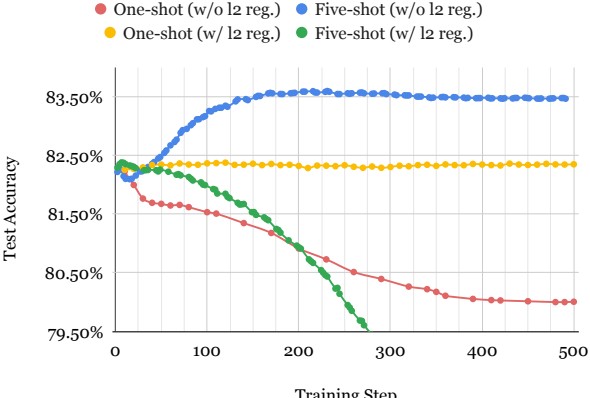

Figure 6: The effect of $L_2$ weight regularization for one-shot and five-shot learning. We plot the top-1 test accuracy of CoCa-base on ImageNet vs. the training step. The $L_2$ weight regularization is beneficial to one-shot learning but harmful to five-shot learning.

finetuned model is easily distorted by the limited training examples, resulting in a degradation from zero-shot performance. However, by applying $L_2$ weight regularization for one-shot learning, the decision boundary does not deviate much from the decision boundary of the pretrained model. This is reflected in the steady increase of test accuracy from 82.26% to 82.35%, as depicted by the yellow curve in Figure 6. Although the performance gain is small, it is still noteworthy since the information provided by one-shot data is limited in helping a pretrained model. On the other hand, applying $L_2$ weight regularization in five-shot learning could adversely affect the model adaptation, as shown by the green curve. The reason is that $L_2$ weight regularization, acting as an additional constraint, restricts the model from learning new knowledge from the training data when sufficient information is available to refine the decision boundary of the pretrained model. It should be noted that all of the aforementioned phenomena are dependent on utilizing category name initialization. The decision boundary will lack discriminative power if category name initialization is not used. Therefore, adding $L_2$ weight regularization would have no meaningful effect.

## 5    Conclusion

This paper has studied the few-shot classification problem using large vision-language models. Since it is hard to optimize large vision-language models with a few training examples, we propose exploring category names to initialize the classification head, significantly improving performance. In addition, we have also

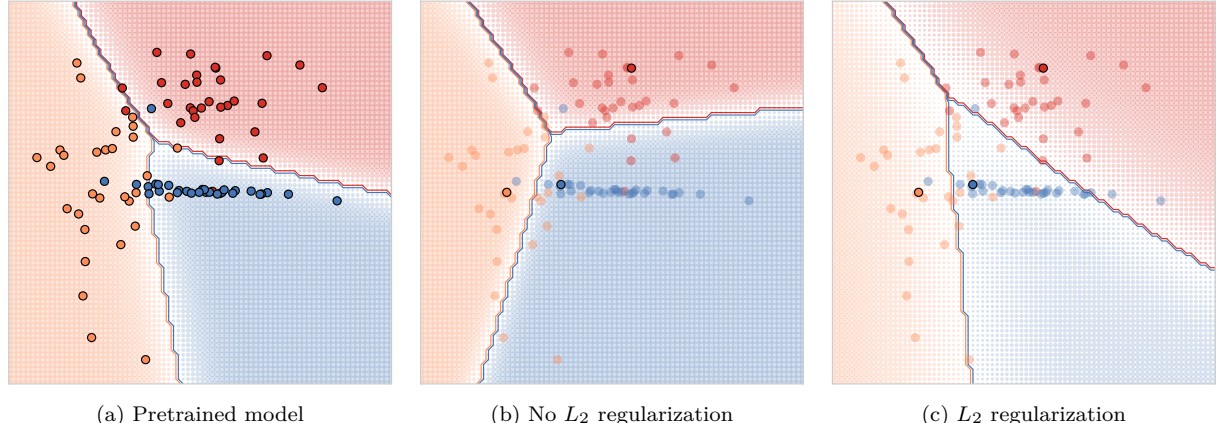

(a) Pretrained model          (b) No $L_2$ regularization          (c) $L_2$ regularization

Figure 7: Visualization of decision boundary in one-shot learning. From left to right, The first subfigure displays the decision boundary of the pretrained model. In contrast, the second and third subfigures show the finetuned model without and with $L_2$ weight regularization, respectively. Each model was trained using only one training example per class, with three classes retained for simplicity. The decision boundary does not shift significantly when finetuning on the one-shot dataset with $L_2$ regularization. This indicates that the model's generalization ability is improved, as it is less likely to overfit the training examples.

investigated the condition when the category names help. We demonstrate that borrowing other non-perfect category names or even names from a foreign language could also help the few-shot classification of vision-language models, which is better than randomly initializing the classification head. However, the contribution of category names diminishes when the number of training samples becomes large. This paper obtains state-of-the-art few-shot performance on numerous benchmarks, including ImageNet, ImageNet-V2, ImageNet-R, ImageNet-A, ImageNet-Sketch, Cifar100, Oxford Flowers, Stanford Cars, Country-211, Food-101, FGVC Aircraft, EuroSAT, and Oxford-IIIT Pets. Our few-shot classification result is even better than many previous works that have employed the whole training set.

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
