# OpenReview forum: "Exploiting Category Names for Few-Shot Classification with Vision-Language Models"
_TMLR — Rejected by TMLR_

### Review · Reviewer_biRF · 2023-05-01

**Summary Of Contributions:**

This article addresses the problem of few-shot image classification using vision-language models. To do so, first, the model is pretraining in a similar way as CLIP. Then, the image classification head is finetuned to predict the classes. Then, it is finetuned again using the weights produced by some promptings of the language models with sentences that contain the actual name of the image classes.

The article clearly states their contributions:
- In the few-shot scenario, the proposed model that initializes the linear projection weights with the prompts' embeddings with the class names improves the SOTA.
- Their model converges quicker than random initialization
- It also works with embeddings from languages other than English
- Other minor contributions

**Audience:**

Yes

**Broader Impact Concerns:**

No ethical concerns.

**Claims And Evidence:**

Yes

**Requested Changes:**

Explain more about hyperparameters and computational cost.

I miss comparison with other SOTA methods like MAPL: Parameter-Efficient Adaptation of Unimodal Pre-Trained Models for Vision-Language Few-Shot Prompting


**Strengths And Weaknesses:**

The article presents a significant contribution to the field of few-shot image classification. The paper proposes a method that leverages the category names to initialize the classification head, which leads to significant improvements in performance. The strengths of the article are numerous.

Firstly, the paper is highly relevant. Few-shot image classification is a challenging task with practical applications in various domains, such as medicine and robotics. The proposed method can significantly improve the performance of vision-language models on this task, which has important implications for real-world scenarios.

Secondly, the article is built on top of state-of-the-art methods. The authors use a large-scale vision-language foundation model and extend it with a novel category name initialization method for the classification head. This approach builds on existing techniques and improves them to achieve better performance.

Thirdly, the paper is evaluated on over 13 different few-shot image classification benchmarks. This extensive evaluation demonstrates the effectiveness and generalizability of the proposed method across various datasets and tasks.

Fourthly, the article is well-written, with a clear and concise presentation of the proposed method, experiments, and results. The authors provide detailed explanations and visualizations, making it easy to understand the proposed approach and its performance.

Fifthly, the results of the proposed method are impressive. The paper shows that the category name initialization method significantly improves the performance of the vision-language model on various few-shot image classification benchmarks. In particular, the proposed approach achieves state-of-the-art performance on ImageNet and Stanford Cars datasets.

Finally, the paper includes meaningful ablation experiments that help to understand the contribution of each component of the proposed method. The authors provide a thorough analysis of the results, demonstrating the effectiveness of the proposed method and providing insights into its limitations.

One weakness of the article is that the proposed method relies on category names to initialize the classification head. While this approach can significantly improve few-shot classification performance, it may not be applicable in all scenarios. For example, in domains where category names are not available or are not reliable, the proposed method may not be effective.

Another potential weakness is that the paper does not provide a detailed analysis of the computational and memory requirements of the proposed method. While the authors mention that their method is efficient, it would be useful to have a more thorough discussion of the computational costs associated with the proposed approach, particularly given the large size of vision-language models.

Additionally, the paper does not explore the impact of hyperparameter choices on the performance of the proposed method. While the authors provide some information on the hyperparameters used in their experiments, a more detailed investigation of the sensitivity of the method to different hyperparameters would be valuable.

---

> ### Author Response · Authors · 2023-06-10
> **Response to Reviewer biRF**
>
> > One weakness of the article is that the proposed method relies on category names to initialize the classification head. While this approach can significantly improve few-shot classification performance, it may not be applicable in all scenarios. For example, in domains where category names are not available or are not reliable, the proposed method may not be effective.
>
> We agree that the proposed category name initialization requires knowing category names. In Section 4.5, we include the update that the method will not be applicable when category names are unavailable.
>
>
> > Explain more about hyperparameters and computational cost.
>
> The choice of batch size depends on the dataset and its number of categories. When the total number of training examples is relatively small, using a large batch size may not be feasible. However, using the largest possible batch size for efficient training is generally desirable. For instance, in the case of ImageNet, which consists of 1000 categories, we opt for a batch size of 512. This decision is based on the consideration that we have a substantial number of images per category, either 1000 images (for one-shot tasks) or 5000 images (for five-shot tasks). Therefore, using a batch size of 512, we can efficiently utilize the available computational resources during training. However, it is important to note that the batch size is adjusted accordingly for datasets with a smaller number of categories. For instance, in the case of Cifar-100, where there are 100 categories, we choose a batch size of 256 for the five-shot setting and 64 for the one-shot setting.
>
> The computational cost of training a model depends on the model size and the chosen training batch size. To provide specific examples, when fine-tuning CoCa-base on ImageNet (five-shot), we utilized a 4x4 Jellyfish TPU with a batch size of 512, and the training process took approximately 6 hours. Similarly, when fine-tuning CoCa-2B on Cifar-100 (five-shot), we employed a 4x4 Dragonfish TPU with a batch size 256, and the training duration was around 9 hours.
>
> The above discussion, the Optimization paragraph in Section 3.2, and the original Section 4.1 (Data) are merged into Section 4.1, renamed Experimental Setup.
>
>
> > I miss comparison with other SOTA methods like MAPL: Parameter-Efficient Adaptation of Unimodal Pre-Trained Models for Vision-Language Few-Shot Prompting
>
> We note that MAPL and our method serve different purposes and target various tasks. MAPL specifically focuses on visual question answering and image captioning benchmarks. On the other hand, our paper aims at image classification tasks.

---

### Review · Reviewer_6zmr · 2023-05-22

**Summary Of Contributions:**

The paper addresses few-shot classification problem using pretrained large Vision Language Models (VLMs). In particular, the authors build their method on top of the Contrastive Captioning (CoCa) [Yu et al., 2022] model, a state-of-the-art VLM. CoCa relies on contrastive objective from vision and unimodal text encoders as well as on captioning objective from the same vision encoder but a multimodal text encoder. After pretraining on web-level noisy image-text pairs, CoCa performs downstream visual (image, video recognition) and multimodal (retrieval, captioning etc.) tasks by finetuning the encoders or training small attention and linear classification layers on top of the frozen encoders. The current work proposes a modification on dealing with the downstream tasks by coming up with a clever initialization mechanism (known as Category Name Initialization or CNI) for the linear classifier. The linear classifier may or may not be trained depending on whether the downstream task is performed zero-shot or not. Given $D$ dimensional output from the encoders and $C$ classes, the linear classifier needs to have $C\times D$ learnable weights. For each of the $C$ classes, the authors propose to obtain $N$ embeddings ($D$ dimensional) by passing each category name along with $N$ different but predefined prompts through the unimodal text encoder. For $C$ category or class names, these will give $C\times D$ matrix which when properly normalized, can act as the initialization weights for the linear classifier. The authors performed evaluations on image classification on different datasets in both zero and few shot scenarios and have shown the superiority of their approach. The authors have also performed good ablation experiments. While the experiments are well thought, a few missing experiments are of concern. As the work is based on top of CoCa, the main competitor seems to be CoOp [Zhou et al., 2022], another school of parameter efficient tuning of VLMs that uses learnable prompts. So, extensive comparison with this is required. I’m detailing it below (in `Requested Changes’).

**Audience:**

Yes

**Broader Impact Concerns:**

The work does not seem to be concerning on the ethical ground. However, a small statement on this would always be helpful.

**Claims And Evidence:**

Yes

**Requested Changes:**

- Table 2: This is where experiments with datasets not specific to CoCa but rather specific to mostly CoOp are performed. A fairer comparison would be to compare CoCa-base and CoCa-2B with random initialization of the task specific attentional pooling on top of the CoCa encoder. In CoCa, this is termed as the Frozen-feature evaluation. This would show the difference it makes to have CNI vis-a-vis random initialization. Though this is done in ImageNet and its variants (Table 3), diverse datasets like the ones examined in CoOp would show the robustness of the approach. CoOp does a fantastic job, by showing their performance on 11 different and diverse datasets. While the proposed work has picked a few of them, the omission of the rest raises the question of how the performance of CNI would be on these missing datasets.
- Category name initialization vs. random initialization (Page 8) and Figure 3: Very little detail is provided here and the associated figure 3. Which dataset is this? Is it for a single dataset only, or average over the 11 datasets on which CoOp does its evaluation? This would be a meaningful analysis only if it is done on these 11 datasets and in ImageNet and its variants.
- CoCa has shown very good performance on video recognition on Kinetics and MiT datasets as well as multimodal classification tasks like VQA or Visual Entailment. Experimenting on these tasks and comparing with related works would increase the applicability of the approach.
- As the main contribution of the paper is in initializing a linear classifier, Xavier initialization [a] can be a good comparison to perform.
- Page 5 says about normalizing to obtain the average embeddings. What sort of normalization is performed is not detailed.
- Writing issues:
  - Introduction, first paragraph last line: 'provides novel perspectives of thinking of few-example learning’ -> 'provides novel perspectives on few-example learning’
  - Page 3: Not sure what is meant by - 'could improve the effectiveness robustness while undermining the relative robustness’.
  - Page 4: '… are training required’ – Not proper usage.
  - Section 3.2: 'the number parameters for’ – ‘of’ is missing.
  - Page 12: 'Figure 5 presents the top-1 test accuracy on ImageNet using different initial learning rates’ – This line is duplicated.
  - Page 3 (Last paragraph, first line): CoOp’s citation is wrong.

[a] Glorot, Xavier, and Yoshua Bengio. "Understanding the difficulty of training deep feedforward neural networks." Proceedings of the thirteenth international conference on artificial intelligence and statistics, 2010.

**Strengths And Weaknesses:**

Strengths:
 - The work provides a new perspective of using information contained in class names for zero/few shot adaptation of large pretrained VLMs towards downstream tasks.
- The number of experiments is large along with well thought and analyzed ablations.

Weaknesses:
- One of the major weaknesses actually is related to the second strength. Though the number of experiments is large, it might not be comprehensive. Many datasets used in CoOp are not evaluated. A related task on which CoCa with random initialization has shown very good performance is video classification. This experiment is completely missing from the proposed approach. Having this would help understand the broad applicability of the approach. Some important CoCa variations on datasets that CoCa has not shown results but CNI shows, are missing.
- The writing is generally good, but I must say things are sloppy at certain places (including duplication of lines).
- Details of some tables/figures are required.
I’m detailing more on the weaknesses in the `Requested Changes’ point below.

---

> ### Author Response · Authors · 2023-06-10
> **Response to Reviewer 6zmr**
>
> > One of the major weaknesses actually is related to the second strength. Though the number of experiments is large, it might not be comprehensive.
>
> Table 3 compares two different initializations, category name, and random initialization, and other fine-tuning methods on ImageNet and its variants. The table below shows their one-shot and five-shot performance on the other eight benchmarks, where RI and CNI denote random initialization and category name initialization. We can observe that the few-shot classification results using random initialization are worse than the zero-shot classification with pretrained CoCa. However, employing category name initialization would effectively use those few training examples and boost performance.
>
> |             | Shot  | Cifar-100 | Oxford Flowers | Stanford Cars | Country-211 | Food-101 | FGVC Aircraft | EuroSAT | Oxford-IIIT Pets |
> | ----------- | ----- | --------- | -------------- | ------------- | ----------- | -------- | ------------- | ------- | ---------------- |
> | CoCa-2B     | 0     | 77.19     | 92.04         | 94.37         | 42.15       | 94.79    | 44.83         | 49.74   | 97.88            |
> | CoCa-2B+CNI | 1     | 77.89     | 98.45    | 95.29         | 42.44       | 94.91    | 58.33         | 75.06   | 97.93            |
> | CoCa-2B+CNI | 5           | 78.62 | 99.25   | 96.08          | 44.52         | 95.5        | 69.29    | 85.78         | 98.12   |
> | CoCa-2B+RI  | 1     | 5.69      | 40.78        | 14.29         | 1.71        | ​​1.26   | 12.24         | 56.84   | 61.95            |
> | CoCa-2B+RI  | 5           | 7.49  | 84.71     | 86.31          | 19.06         | 62.45       | 27.21    | 82.38         | 78.61   |
>
> CoOp has shown its performance in 11 different datasets (including ImageNet). In Table 2, we have demonstrated our performance in 13 datasets (including ImageNet and its variants), which is not a few. In contrast, our evaluation datasets have a considerable overlap with CoOp’s. On those common evaluation benchmarks, our method achieves better few-shot performance than CoOp, such as ImageNet (see Table 3), OxfordPets (ours: 98.12% vs. CoOp 85%), Stanford Cars (ours: 96.08% vs. 65%), Oxford Flowers (ours: 99.25 vs. CoOp 89%), Food-101 (ours: 95.50% vs. CoOp 72%), FGVC Aircraft (ours: 69.29% vs. CoOp 25%), and EuroSAT (ours: 85.78% vs. CoOp: 73%).
>
> We update our comparison in Table 2 and discussion in the paragraph on Category name initialization vs. random initialization.
>
> > Category name initialization vs. random initialization (Page 8) and Figure 3: Very little detail is provided here and the associated figure 3.
>
> Figure 3 shows the few-shot test accuracy on ImageNet. We chose ImageNet because it is the most complicated and representative dataset with the largest number of categories among all the evaluated datasets. Employing category name initialization for other datasets will also help the model converge better and faster. For example, we plot the test accuracy curves along the finetuning process on Cifar-100 below. We include more examples in Figure 3. Please check out our revised paper or the following picture links to view images.
>
> [Random Initialization](https://imgtr.ee/images/2023/06/10/KJTw4.png)
>
> [Category Name Initialization](https://imgtr.ee/images/2023/06/10/KA6Ql.png)
>
>
> > Experimenting on these tasks and comparing with related works would increase the applicability of the approach.
>
> This paper focuses on category name initialization for few-shot image classification tasks, while Kinetics/VQA/Visual Entailment will require extra modeling efforts. We leave an extension to these tasks in future work.
>
> > Xavier initialization [a] can be a good comparison to perform.
>
> The random initialization in this paper means that the weight values are sampled from a Gaussian distribution N(0, 0.01). Xavier initialization also denotes that the weight values are sampled from a Gaussian distribution but with a different variance, 1/(N_in + N_out), where N_in and N_out specify the input and output dimensions. The results of using Xavier initialization are similar to those of random initialization. Without category name initialization, the model’s few-shot performance at the initial step would be near 0.
>
>
> >  What sort of normalization is performed is not detailed.
>
> We perform the L2 normalization to the averaged text embedding tensor, the same as the previous work CLIP. We add the clarification in the second paragraph of Section 3.2.
>
> > Writing issues
>
> As suggested, we have fixed the typos in the revised manuscript.
>
> The definition of effective robustness and relative robustness can be found in (Radford et al., 2021). Effective robustness measures improvements in accuracy under distribution shift above what is predicted by the documented relationship between in-distribution and out-of-distribution accuracy. Relative robustness captures any improvement in out-of-distribution accuracy.

---

### Review · Reviewer_U7zf · 2023-05-28

**Summary Of Contributions:**

The following work evaluates the effectiveness of using pretrained
image and text embeddings as initialization weights for training few
or zero shot classifiers. Authors evaluate the effectiveness of the
initialization setup with respect to recent advances in large-scale
pretrained LLMs. The approach is compared against standard
initialization schemes that do not leverage linguistic
structure. Further ablations are evaluate performance under the
multi-lingual scenario as well as diminishing returns when additional data is available.

**Audience:**

Yes

**Broader Impact Concerns:**

Authors did not include a broader impact statement

**Claims And Evidence:**

Yes

**Requested Changes:**

- Include discussion for prior embeddings-as-classifiers approaches from
older vision-language literature.
- Address concerns regarding potential overlap between pretraining
data and test data

**Strengths And Weaknesses:**

Strengths:
+ detailed hyperparameter sweep as well as identifying point of
diminishing returns
+ Principled approach to leverage pretrained multimodal embedding
models
+ perhaps a useful revisiting of a fundamental technique in
vision-language literature in the context of more recent advances
+ highlights a pretty notable issue in the initialization approach of
the prior work they build upon

Weaknesses:
- As previously mentioned, embeddings-as-classifiers [3] has been a
  standard approach in vision-language literature since (and possibly before)
  introduction of word2vec. Authors should include relevant literature
  from phrase-grounding [1] works. There also
  exist prior works with the same or similar category name initialization
  setup such as [2]
- While the multilingual experiments are interesting, I think it would
  be pretty straightforward to run a translator to map everything to
  english, effectively eliminating performance gap?
- Another potential issue that applies to both this work and its
  predecessor is that of dataset leakage. I did not find any
  discussion regarding potential overlap of pretraining data and test
  data in either this manuscript or that of CoCa. While the approach
  itself is principled, any leakage would invalidate the reported
  numbers.

[1] Wang et al. Learning Two-Branch Neural Networks for
Image-Text Matching Tasks. PAMI 2018

[2] T Gupta, K Shih, S Singh, D Hoiem. Aligned Image-Word Representations Improve Inductive Transfer Across Vision-Language Tasks. ICCV 2017

[3] Klein, B., Lev, G., Sadeh, G., and Wolf, L. (2014). Fisher
vectors derived from hybrid gaussian-laplacian mixture
models for image annotation

---

> ### Author Response · Authors · 2023-06-10
> **Response to Reviewer U7zf**
>
>
> > As previously mentioned, embeddings-as-classifiers [3] has been a standard approach in vision-language literature since (and possibly before) introduction of word2vec. Authors should include relevant literature from phrase-grounding [1] works. There also exist prior works with the same or similar category name initialization setup such as [2]
>
> > Include discussion for prior embeddings-as-classifiers approaches from older vision-language literature.
>
> As suggested, we include the following discussion in the last paragraph of Section 2.
>
> Klein et al. (2014) suggest that using a fisher vector derived from other distributions can improve accuracy in central computer vision tasks. Category names have also been exploited in image-text tasks, such as visual grounding (Wang et al., 2017), and visual question answering  (Gupta et al., 2017). In these methods, the text embedding of the category names and the image embedding is extracted separately by two branches. Then their inner product is calculated as the similarity score between an image region and an object category.
>
> > While the multilingual experiments are interesting, I think it would be pretty straightforward to run a translator to map everything to english, effectively eliminating performance gap?
>
> Utilizing a translator to convert everything into English would be effective in practical applications when resources are not an issue. We emphasize that this work provides new research insights and effective solutions when the resources are limited (and we cannot afford an extra translator).
>
> > Another potential issue that applies to both this work and its predecessor is that of dataset leakage. I did not find any discussion regarding potential overlap of pretraining data and test data in either this manuscript or that of CoCa. While the approach itself is principled, any leakage would invalidate the reported numbers.
>
> > Address concerns regarding potential overlap between pretraining data and test data.
>
> We note that CoCa (Yu et al., 2022) is pretrained using JFT- 3B (Zhai et al., 2021) and Align datasets (Jia et al., 2021). In the pretraining stage, all near-domain examples (3.6M images) are removed following the strict de-duplication procedures (Zhai et al., 2021; Jia et al., 2021). Please check the first paragraph in Section 4.1 of the CoCa paper for detail.
>
> We have updated the discussion in Section 4.1 of our revised manuscript.
>
> References:
>
> - Xiaohua Zhai, Alexander Kolesnikov, Neil Houlsby, and Lucas Beyer. Scaling vision transformers. CVPR, 2022
>
> - Chao Jia, Yinfei Yang, Ye Xia, Yi-Ting Chen, Zarana Parekh, Hieu Pham, Quoc Le, Yun-Hsuan Sung, Zhen Li, and Tom Duerig. Scaling Up Visual and Vision-Language Representation Learning With Noisy Text Supervision, ICML, 2021
>
> - Jiahui Yu, Zirui Wang, Vijay Vasudevan, Legg Yeung, Mojtaba Seyedhosseini, and Yonghui Wu. Coca: Contrastive captioners are image-text foundation models. TMLR, 2022

---

### Decision · Action_Editors · 2023-07-10

**Recommendation:** Reject

**Comment:**

Thanks for your submission.  This paper was borderline, with two reviewers leaning accept and one leaning reject.  From my read of the reviews, it seems that there are still some unaddressed concerns, particularly from reviewer 6zmr.   In particular, after the rebuttal period this reviewer feels that there are two important experiments missing.  I'm going to paste in a comment that the reviewer made to me after the rebuttal period which was not visible to you, as it neatly summarizes the issues that this reviewer has with the paper as is:

"As I said in my review CoOp [Zhou et al., 2022] being in another school of parameter efficient tuning that uses learnable prompts, is one of the crucial competitors. In five important datasets (Caltech101, Flowers102, SUN397, DTD, UCF101) where CoOp showed their performance, the proposed approach did not show any comparison as I asked in the review. However, they have shown comparison on 6 selective datasets out of the total 11. The missing experiments are important to get a wholesome view of the usefulness of the proposed approach as one of them is unique in the sense that it is a video dataset while another is a texture dataset. The second missing experiment is related to the main approach (CoCa) on which it is built on. CoCa has shown very good performance on video recognition and other multimodal classification as well. As I told in my review, these are also classification tasks and falls in the purview of comparison with the proposed classification approach. With the computation effort mentioned as a response to one of the fellow reviewers, I think, these experiments could have been performed. In summary, though the work has shown many experiments, with the missing experiments it seems to me as lacking the completeness to be accepted in TMLR in its current form."

These seem to be the main concerns, as the other reviewers have indicated that they're happy with the rebuttal provided by the authors.  I would like to see these concerns addressed in a major revision to the paper, with another round of reviewing, before having this paper accepted to TMLR.  I think addressing these concerns will make the paper considerably stronger.

**Audience:**

Yes, this is a paper that would be relevant to the audience.

**Claims And Evidence:**

There are a number of experiments in the paper, but one of the reviewers feels that there could be more evidence presented and that the paper is not ready in its current form.  See comments below for details on the requested experiments that the authors still need to provide.

**Resubmission Of Major Revision:**

The authors may consider submitting a major revision at a later time.